# Kidney and Liver Tissue Tacrolimus Concentrations in Adult Transplant Recipients—The Influence of the Whole Blood and Tissue Concentrations on Efficiency of Treatment during Immunosuppressive Therapy

**DOI:** 10.3390/pharmaceutics13101576

**Published:** 2021-09-28

**Authors:** Magdalena Bodnar-Broniarczyk, Magdalena Durlik, Teresa Bączkowska, Katarzyna Czerwińska, Ryszard Marszałek, Tomasz Pawiński

**Affiliations:** 1Department of Drug Chemistry, Medical University of Warsaw, Banacha 1, 02-097 Warsaw, Poland; mbroniarczyk@wum.edu.pl; 2Department of Transplantation Medicine, Nephrology and Internal Medicine, Medical University of Warsaw, Nowogrodzka 59, 02-006 Warsaw, Poland; magdalena.durlik@wum.edu.pl (M.D.); tbaczkowska@wum.edu.pl (T.B.); kczerwinska2@wum.edu.pl (K.C.); 3Department of Drug Bioanalysis and Analysis, Medical University of Warsaw, Banacha 1, 02-097 Warsaw, Poland; ryszard.marszalek@wum.edu.pl

**Keywords:** tacrolimus, kidney and liver transplantation, tissue concentration, LC-MS/MS, TDM

## Abstract

Tacrolimus (TAC) has a narrow therapeutic index and highly variable pharmacokinetic characteristics. Close monitoring of the TAC concentrations is required in order to avoid the risk of acute rejection or adverse drug reaction. The results in some studies indicate that inter-tissue TAC concentrations can be a better predictor with regards to acute rejection episode than TAC concentration in whole blood. Therefore, the aim of the study was to assess the correlation between dosage, blood, hepatic and kidney tissue concentration of TAC measured by a validated liquid chromatography tandem mass spectrometry (LC-MS/MS) and clinical outcomes in a larger cohort of 100 liver and renal adult transplant recipients. Dried biopsies were weighed, mechanically homogenized and then the samples were treated with a mixture of zinc sulfate—acetonitrile to perform protein precipitation. After centrifugation, the extraction with *tert*-butyl methyl ether was performed. The analytical range was proven for TAC tissue concentrations of 10–400 pg/mg. The accuracy and precision fell within the acceptance criteria for intraday as well as interday assay. There was no correlation between dosage, blood (C_0_) and tissue TAC concentrations. TAC concentrations determined in liver and kidney biopsies ranged from 8.5 pg/mg up to 160.0 pg/mg and from 7.1 pg/mg up to 215.7 pg/mg, respectively. To the best of our knowledge, this is the first LC-MS/MS method for kidney and liver tissue TAC monitoring using Tac^13^C,D_2_ as the internal standard, which permits measuring tissue TAC concentrations as low as 10 pg/mg.

## 1. Introduction

Therapeutic Drug Monitoring (TDM) of immunosuppressive agents is an integral part of effective pharmacotherapy of patients after organ transplantation [1]. Immunosuppressive agents are critical dose drugs exhibiting desirable therapeutic effects with an acceptable tolerability within a narrow range of blood concentrations. The correlation between blood drug concentrations and clinical outcomes is an important factor supporting the use of TDM. Despite plasma and whole blood being used in routine TDM as the most popular matrices, they have a number of limitations [2]: venous blood sampling, centrifugation and shipment at low temperatures. Furthermore, drug concentrations in plasma or blood rarely reflect drug or metabolite level in tissues or cells.

Tacrolimus (TAC), an inhibitor of calcineurin, has been used widely to prevent organ rejection for a quarter of a century [3]. TAC exhibits wide inter-individual as well as intra-individual pharmacokinetics variability in both kidney and especially liver transplant recipients [4]. In the kidney transplant group, the dosage alone failed to correlate with total drug exposure, whereas there was a linear relationship between trough blood concentrations of TAC and both the maximum concentration and the area under the concentration-time curve (AUC) during the dosing interval. This fact indicated that trough blood TAC concentrations measured in kidney transplant recipients could be used as a surrogate measures of total drug exposure during the dosing interval and might, therefore, serve as an intermediate therapeutic end-point for guiding dosage adjustment. For the last several years, the correlation between individual TAC concentrations and AUC_0–12_ has been studied in kidney and liver transplant recipients [5]. The troughs’ TAC level can usually be used as a surrogate for exposure, but tissue distribution can be influenced by factors disturbing the extracellular–intracellular equilibrium, potentially resulting in unpredictable tissue concentrations. This change in tissue concentrations might then preclude ensuring drug efficacy and adverse drug reaction onset.

In the last decade, evidence suggests that intragraft and intracellular TAC concentrations not only among immunosuppressive agents may more accurately predict transplant outcomes [6,7,8,9,10]. Tissue TAC concentrations within the allograft might provide a better understanding of TAC distribution during graft rejection. According to the updated Second Consensus Report [3], monitoring TAC concentrations in tissue was classified as a new monitoring strategy. It may provide a better understanding of TAC distribution mechanisms during graft rejection. Measuring TAC directly in graft tissue may be of interest as it is justifiable to expect that local concentrations show drug effects more precisely. Interest in such approaches has been reinforced by observations of the weak relationship between intracellular and whole blood TAC concentrations in patients of different types of transplantation, suggesting potential added value.

The aim of the study was to assess the correlation between whole blood, hepatic and kidney tissue concentrations of TAC measured by a validated LC-MS/MS and clinical outcome in a cohort of 100 liver and renal transplant recipients. During the investigation, the authors tried to define the influence of blood and tissue TAC concentration level on efficiency of treatment in immunosuppressive therapy.

## 2. Materials and Methods

### 2.1. Patients, Population and Immunosuppression Protocol

One hundred adult liver and kidney (73 men vs. 27 women; median age, 46.3) transplant patients were routinely followed between October 2014 and December 2018 in the Department of Transplantation Medicine, Nephrology and Internal Medicine Medical University of Warsaw. Patients provided written informed consent before being included in the study. The study was approved by our local ethics committee (Medical University of Warsaw, Warsaw, Poland, KB/58/2012).

All patients were administered with TAC (Prograf) and mycophenolate immunosuppression (CellCept, mycophenolate mofetil), and 50 of these patients received low-dose steroids immunosuppression using Encorton. The subjects were not receiving potentially interacting drugs.

Blood samples were drawn, and biopsies were performed on the same day: blood sample at 7:00 a.m. and biopsy at 10:00 a.m.–11:00 a.m. For the allograft tissue samples, both the kidney and the liver were the same samples also used for histology assessment. Acute rejection was graded in the case of kidney biopsy according to the BANFF score and according to the RAI index (Rejection Activity Index) in the case of liver biopsy.

Drug-free homogenates were obtained from human kidneys in conjunction with nephrectomy due to neoplastic changes (tumor). Drug-free homogenates were obtained from a human liver from a patient with indication for transplantation due to HCC (hepatocellular carcinoma) without cirrhosis. The patients had never been treated with TAC. The method development protocol and the clinical trial protocol were evaluated and approved by the Regional Bioethical Committee and performed in accordance with the Declaration of Helsinki. All patients signed a written informed consent.

### 2.2. Chemicals and Reagents

Tacrolimus was a gift from Astellas Pharma Inc. (Tokyo, Japan), and stable isotope-labeled tacrolimus (Tac^13^C,D_2_) was purchased from Toronto Research Chemicals Inc. (Toronto, ON, Canada). The reagents for chromatography are as follows: hyper LC/MS grade methanol and water, as well as zinc sulfate heptahydrate, were purchased from Merck Millipore (Darmstadt, Germany). Hyper LC/MS-grade acetonitrile, formic acid and HPLC-grade tert-butyl methyl ether were obtained from J.T. Baker (Deventer, the Netherlands), whereas ammonium acetate Optigrade was purchased LGC Standards (Wesel, Germany). Reagent-grade deionized water was produced using a Millipore SimPak1, Simplicity 185 (Millipore, France).

### 2.3. Calibration Standards and Quality Control Samples

Calibration curves (n = 15) included a blank sample (drug-free homogenate sample processed without internal standard IS), a zero sample (drug-free homogenate sample processed with IS) and six spiked samples (nonzero homogenate samples) covering the range from 10 to 400 pg/mg of kidney or liver tissue. Routine daily calibration curves and controls were prepared for each analytical batch in drug-free homogenate of 0.9 mg to 3.8 mg of human kidney or hepatic tissue in phosphate buffer. Calibrators were made to reach concentrations of 10, 50, 100, 200, 300 and 400 pg of TAC per milligram of dry tissue. The quality control (QC) samples were prepared in the same way at three levels: 20, 150 and 250 pg/mg. The stock solution of TAC was prepared by dissolving an accurately weighed amount of drug in methanol, resulting in concentrations of 1 mg/L. A 1 mg/L stock solution of Tac^13^C,D_2_ used as IS in methanol was prepared and further diluted to reach a 0.1 µg/mL working solution. All stock solutions were stored at −80 °C and were stable for at least 1 month.

Our laboratory participated in the TAC International Proficiency Testing Scheme organized by the Analytical Service International (ASI, Ltd., St. George’s University of London, Cranmer Terrace, London, UK) and next after 2017 by LGC Limited, LGC Standards PT, Immunosuppressant (Teddington, UK). Whole blood tacrolimus concentrations were measured using a validated LC-MS/MS method developed in our laboratory [11].

### 2.4. TAC Extraction from Kidney and Liver Tissue Samples

Transplant biopsies (0.9 to 3.8 mg) were stored in oval tubes (Simport) and deep-frozen at −80 °C until analysis. At the time of analysis, the specimens after defrosting were accurately weighed and homogenized using Heidolph homogenizer in 0.7 mL 5 mmol/L tris buffer pH 5.5 (kidney) and 7.0 (liver) using Eppendorf micropestle tubes (30 s 75,000 rot/min). Right after homogenization, the internal standard was added. To precipitate protein, 1 mL mixture of 0.1 mol/L zinc sulfate—acetonitrile (50:50, *v/v*) was used. The mixture was then vigorously rotary mixed for 20 min at room temperature. After centrifugation (3000 rpm for 10 min at 5 °C), the supernatant was quantitatively transferred into the glass tube, and the extraction with 3.0 mL tert-butyl methyl ether using a rotary mixer (30 min at 1200 rpm) was performed next. Subsequently, sample centrifugation at 3000 rpm for 10 min at 5 °C was performed. After that, the organic layer was transferred into another glass tube, and it evaporated to dryness under a stream of nitrogen in a thermostatically controlled and maintained water bath.

### 2.5. LC-MS/MS System and Conditions

Ten microliters of sample was injected into an Agilent 1260 Infinity liquid chromatography system (Agilent Technologies, Palo Alto, CA, USA) consisting of a binary pump, degasser, thermostated column compartment, autosampler and thermostat for the autosampler. Analytes were eluted into the mass spectrometer via Poroshell 120 EC-C18 (4.6 × 50 mm, 2.7 µm), which was maintained at 50 °C and guarded with Poroshell 120 EC-C18 (4.6 × 5 mm, 2.7 µm) precolumn, and both are from Agilent Technologies. The mobile phase pumped in a gradient mode consisted of a mixture of solvents A and B. Solvent A was 2.5 mmol/L ammonium acetate and 0.1% formic acid in water. Solvent B was 2.5 mmol/L ammonium acetate and 0.1% formic acid in methanol. A binary step gradient at a flow rate of 0.75 mL/min was employed. The gradient program was as follows: 90% of solution A and 10% of solution B from the start of analysis until 2.0 min and then changed to 5% of solution A and 95% of solution B from 2.0 min to 6.0 min. At 6.1 min, the mobile phase was reverted back to 90% of solution A and 10% of solution B. The autosampler temperature was maintained at 4 °C.

The Analyst 1.6.1 software (AB Sciex, Concord, ON, Canada) was used for peak area counting, calibration fitting, TAC concentrations calculating and also signal-to-noise ratio determining.

Analyte detection was achieved by using positive electrospray ionization (ESI) with a 4000 QTRAP triple quadrupole mass spectrometer (AB Sciex, Concord, ON, Canada). The ammonium adduct of each analyte [M + NH_4_]^+^ was monitored with mass transitions of 821.5 → 768.4 m/z and 824.6 → 771.5 m/z for TAC and TAC^13^C,D_2_, respectively. A dwell time of 150 ms was used for each mass transition. Nitrogen was used as both the curtain and collision gas. Collision energy was set to 31 V, and an entrance potential of 10 V and the declustering potential of 96 V for TAC and 91 V for IS are used. Ion source parameters were as follows: an ESI voltage of 4.5 kV, a desolvation temperature of 400 °C, the curtain gas of 20, GS1 of 50 and GS2 of 60 units. The representative chromatograms of TAC samples and IS are shown in Figure 1.

### 2.6. Method Validation

Method validation was conducted based on the European Medicines Agency (2011) guidelines. The following method characteristics were evaluated: selectivity, matrix effect, calibration and linearity, accuracy and precision and stability. The analyst 1.6.1 software (AB Sciex, Concord, ON, Canada) was used for peak area integration, calculation of calibration line and TAC concentrations and determination of signal-to-noise ratio directly from the chromatograms.

#### 2.6.1. Matrix Effect

The matrix effect was determined by using the post-column infusion method and post-extraction addition. The method was performed by infusing 20 pg/mg and 200 pg/mg MPA, both coming from kidney and liver tissues, and injecting them into the sample under the established chromatographic conditions. This method enabled the investigation of matrix effects over the entire chromatographic run.

#### 2.6.2. Autosampler Stability

Autosampler stability was measured using three replicates for each TAC concentration level (low QC 20 pg/mg and high QC 250 pg/mg). The samples were evaluated immediately after preparation and subsequently after 4, 10, 16 and 24 h of sample storage in the autosampler rack at 4 °C.

#### 2.6.3. Short-Term Stability

Low QC 20 pg/mg and high QC 250 pg/mg tissue TAC samples were analyzed for testing postpreparation short-term stability. Spiked TAC samples were evaluated immediately after sample preparation (standard analytical procedure, n = 3).

#### 2.6.4. Working Solution Stability

Working solutions of TAC prepared as described in Section 2.3 were stored in a freezer at −80 °C. They were used to prepare calibration standards and QC samples for each analytical run, and they were refrozen immediately after use. Therefore, the stability of working solutions with concentrations that fell within the analytical range of the method (20 pg/mg and 150 pg/mg) was assessed during a 5-week period.

### 2.7. Data Analysis

All data are presented as mean ± standard deviations (SD), and CVs were calculated as SD/mean and expressed as a percentage. The relationships between blood and tissue concentrations were assessed by Spearman’s correlation or linear regression analysis using Prism (GraphPad Software, Inc., La Jolla, CA, USA).

## 3. Results

### 3.1. Assay Validation

The validation of the method was performed according to European Medicines Agency (EMA) guidelines [12,13,14]. Validation parameters, i.e., selectivity, accuracy and precision, range with lower limit of quantification (LLOQ), matrix effect, calibration and linearity, were evaluated independently for TAC concentrations in kidney and liver tissues.

Biopsy-sized TAC-free tissues were sourced from a human kidney and liver biopsy from a transplant patient on TAC-free immunosuppression scheme. The signals from endogenous or unknown substances were <15% of the LLOQ value (1.5 pg/mg) for TAC. Specificity of the assay was assessed by fortifying five blank homogenate samples of 1 mg kidney and liver tissues obtained from different patients without TAC treated with other immunosuppressive agents, cephalosporins, aminoglycosides, benzodiazepines and antiviral agents.

The linearity of the method was evaluated for TAC kidney or liver tissue concentration in the range between 10 and 400 pg/mg from a set of 15 calibration curves. Each curve consisted of standards at 6 levels. A weighted (1/x) linear regression was used in constructing the calibration curve to ensure optimal fitting at low TAC concentrations. Calibration lines were characterized by satisfying values of a coefficient of determination: r^2^ = 0.9993 ± 0.0004 for kidney TAC tissue concentration and r^2^ = 0.9997 ± 0.0002 for liver TAC tissue concentration.

Accuracy and precision were evaluated both in within-run as well as between-run experiments.

LLOQ value was experimentally set at 10 pg/mg. The required level of accuracy and precision was obtained for IS used. The QC’s intra-assay accuracy was measured at 96.5–102.4% and 97.9–105.1% for TAC in kidney and liver tissues, respectively, while inter-assay accuracy was measured at 102.4–111.5% and 93.4–107.9% for concentrations of TAC in kidney and liver tissues. The QC’s intra-assay precision was calculated at 2.1–6.7% and 1.9–6.7% for TAC in kidney and liver tissues, respectively, while inter-assay precision was found at 2.5–9.3% and 8.2–10.5% for TAC in kidney and liver tissues, respectively. The intra-assy and inter-assay precision and accuracy of the LC-MS/MS method for TAC QC samples in kidney and liver tissues at different tissue concentrations and matrix effects are summarized in Table 1.

Two independent methods, postcolumn infusion and postextraction addition, were evaluated for qualitative and quantitative matrix effects, testing similarly to those presented in our previous paper [14]. The matrix effects were acceptable according to the EMA guideline. The results of this experiment confirm that matrix effects have a minimal influence on the method.

Stability (autosampler, short-term and working solution) was tested similarly to the procedure presented in an earlier paper [11]. The autosampler stability was satisfactory. After 24 h of storage in the autosampler rack at 4 °C, the stability of low QC and high QC amounted to 98.78% and 100.57% of the initial value, respectively. The results also proved satisfactory short-term stability. Stability before the preparation procedure was observed at the level of 97.91% and 98.21% for low QC and high QC, respectively. The stability of samples resting at ambient temperature after preparation procedure was also confirmed (95.74% and 104.65% for low QC and high QC, respectively).

The TAC working solutions were stable during entire observation time. The back-calculated concentrations fell within ±15% of the initial value and amounted to 102.34% and 94.56% during the fifth week for solutions at 20 pg/mg and 150 pg/mg, respectively.

### 3.2. Clinical Data

Clinical data for the transplant recipients are shown in Table 2.

The blood and tissue samples were collected between 7 and 180 days post transplantation. TAC concentrations found in kidney and liver biopsies ranged from 7.1 pg/mg to 215.7 pg/mg and from 8.5 pg/mg to 160.0 pg/mg, respectively. It displayed mean (±SD) values of 55.9 ± 33.6 pg/mg (liver transplant recipients) and 65.2 ± 50.0 pg/mg (kidney transplant recipients). Simultaneously mean TAC blood concentrations ranged from 2.0 ng/mL to 14.5 ng/mL (kidney transplant patients) and 2.3 ng/mL to 14.9 ng/mL (liver transplant patients) (see: Table 2). No relationship has been observed between dose and TAC tissue levels.

Figure 2A–D shows the correlation between tacrolimus tissue concentrations and daily dose (A,C) or blood C_0_ (B,D) in liver and kidney transplant recipients, respectively. As shown in Figure 2, there were no significant correlations between kidney tissue concentration, dose and C_0_ blood. Acute rejection episodes for transplant recipients (graft: liver n = 6, kidney = 7) have been indicated in Figure 2 by using empty red triangle.

When compared with the BANFF scores (0–9) of tissue TAC levels in transplant patients with no or mild cellular rejection (score < 6, n = 43), the mean value of renal tissue concentration was 88.9 ± 41.3 pg/mg, whereas the mean value was 23.1 ± 8.1 pg/mg (*p* = 0.2930, ns) in patients with moderate to severe rejection (score > 6, n = 7). The blood concentrations of TAC in both group of recipients were comparable (7.1 ± 1.9 ng/mL vs. 7.8 ± 2.1 ng/mL). In the case of liver transplant patients, the mean value TAC concentration was 28.3 ± 7.8 pg/mg (n = 6) in comparing hepatic tissue Tac levels with Rejection Activity Index (RAI) > 6 providing moderate to severe rejection, while hepatic TAC tissue levels of patients with none or mild rejection (RAI 2–5) were equal 109.2 ± 45.7 pg/mg (n = 44) (*p* = 0.3469). Simultaneously, whole blood TAC concentrations in both group of patients with or without rejection episodes were statistically significant (6.9 ± 1.7 ng/mL vs. 7.4 ± 2.3 ng/mL) (*p* < 0.001). Moreover, the relationship between low TAC concentrations and BANFF score > 6 was statistically significant (for liver transplant patients, *p* = 0.0069; for kidney transplant patients, *p* = 0.0021).

Figure 3 and Figure 4 shows kidney and liver tissue and blood TAC levels determined in 100 transplant patients.

## 4. Discussion

From the point of view of the site of action, the ideal matrix for TDM of immunosuppressive agents would be the target tissue of transplanted organs or target cell: lymphocytes and peripheral blood mononuclear cells (PBMC). However, due to several reasons, we are still very far from adopting them into routine clinical practice. The lack of standardization of data, the variability in cellularity of biopsies, the impact of matrix effect and blood contamination are among reasons for not adopting them in clinical practice. In several investigations, intragraft and intracellular TAC concentrations were suggested to predict transplant outcomes better [5,6,8,9,10,15,16,17,18,19,20,21,22]. Moreover, tissue TAC concentrations within the allograft might provide understanding of TAC distribution during graft rejection more precisely.

Therapeutic drug monitoring of immunosuppressive agents requires precise and accurate analytical methods, especially for the lower ranges of concentrations. Several approaches have been proposed for an LC-MS/MS procedure for this purpose [23,24,25,26]. The LC-MS/MS method displayed precision and accuracy suitable for application to Tacrolimus measurements in human kidney and liver biopsy tissues. In discussing the developed and validated LC-MS/MS method, it has to be noted that the chromatographic conditions were adjusted in such a manner that TAC retention time appears to be relatively short, and isotope-labelled TAC^13^C,D_2_ are applied as the IS assures repeatability of results. Moreover, calibration standards were properly adjusted with regards to therapeutic concentration ranges in tissue.

According to our knowledge, this is one of the first studies to compare the relationship between tacrolimus dose, trough blood and tissue tacrolimus concentrations both in kidney and liver grafts in such a large group of patients including clinical cases of significant organ rejection. The most frequently used means of TAC monitoring is predose trough concentration (C_0_) measurements in whole blood. The most frequently used means of TAC monitoring—predose trough concentration as a pharmacokinetic parameter—was performed along with drug determination in graft tissues. It was supposed to answer the question of whether TAC concentration in tissues can render treatments more efficient.

In contrast to recently published results [27], we have observed no significant correlation between daily doses of tacrolimus and graft concentration (*p* = 0.457 and *p* = 0.424, respectively). Similarly, a weak correlation between C_0_ blood and tissue concentration was displayed both in kidney and in liver organs (*p* = 0.151; *p* = 0.391). Explaining the possible mechanism or reason for the lack of correlations is not easy as the cause is multifactorial. Patients expressing CYP3A5 require at least 50% higher TAC doses to reach the target therapeutic range compared with non-expressors [3]. According to current knowledge about some aspects of pharmacogenetics during immunosuppressive therapy, the association between CYP3A5 genotype and TAC dose requirements is consistent and has been observed among kidney and liver transplant recipients. On the other hand, distribution of TAC to transplanted tissue would be closely modulated by the expression and polymorphisms of drug transporters, e.g., ABCB1 polymorphisms could be partly responsible for the wide range of TAC concentrations observed in graft tissue. It may have influence on the lack of correlation between blood and tissue TAC concentrations. Moreover, due to the time difference between the administered drug and the biopsy performed (3–4 h), which was the case in our study, the correlation between dose and tissue concentration was invisible.

Comparing our results with the previous ones published by Capron A. et al. [5], Noll B.D. et al. [10], Krogstad V. et al. [28] and Sallustio B.C. et al. [19], the following conclusion has to be stated. A similar concentration range was observed in the tissues of patients: 5–387 pg/mg in liver tissue of 146 patients, and the tissue TAC concentration is less than 30 pg/mg. It was a cut-off point to discriminate clinically significant cellular rejection [5]: 119–285 pg/mg in kidney biopsies of two patients, and TAC concentrations were measured over 16–300 days post-transplantation [10]; 43.7 pg/mg and 62.6 pg/mg in renal core biopsies in only two patients [28]; and 33–828 pg/mg in renal biopsies from 132 renal transplant recipients [27].

The aim of the investigation was to define correlations between dose, whole blood, hepatic and kidney tissue concentrations of TAC in the case of our study. On the other hand, other studies’ primary goals were the following: confirmation of acute nephrotoxicity associated with TAC concentration in renal tissue and that acute nephrotoxicity depended on one very high graft concentration (828 pg/mg) [19]; assurance measurement of P-gp expression and of the demethylated metabolites of TAC in the same renal biopsy using the validated LC-MS/MS method for quantification of TAC in tissue homogenates [20]; and, finally, the development of a method for the quantification of TAC in small biopsy-sized samples of rat kidney and liver tissues [10].

By assessing the results obtained in this study with previous ones sparingly found in the literature, excluding cases of isolation from other matrices [21,22,23,24,25,26,27,28], it has to be stated that a low value of TAC concentrations in the tissue of transplanted organs may favor the risk of occurrence the acute rejection episodes. We are convinced that other matrices used in TAC concentration determination could be more complicated, for example, bile [29] or urine. There is no doubt TAC PBMCs level as a marker of efficiency early after transplantation could be an important factor for optimizing immunosuppression after liver and kidney transplantation [17,19]. A study confirming the link between intracellular TAC concentration and patient outcomes after liver transplantation suggested that this new TDM approach was a valuable option, but definitive clinical verification and validation remain to be generated [17]. Moreover, in that case, daily TAC monitoring was performed using CMIA on the ARCHITECT platform, and TAC measurements in tissues were performed by LC-MS/MS. A more recent study [19] confirms that intra-cellular immunosuppressive drug monitoring states are additional tools for more precisely individualizing early immunosuppressive schemes after liver transplantation. Its clinical application appears easier than the tissue drug measurement, which requires invasive biopsies. On the other hand, it cannot be a routine procedure for TAC measurement, and it still needs to be evaluated in longer prospective trials where genetic aspects and different immunosuppression schemes have to be considered.

The limitation of our study was undoubtedly the lack of research applied on the potential contribution of the donor and the recipient CYP3A5 and ABCB1 genetic polymorphisms on graft tacrolimus concentrations. Secondly, the retrospective design may limit the accuracy of the data with possible missing or misreported information from the electronic medical record. There was also the inability to determine the appropriateness of TAC trough concentrations assessed. Co-administration of TAC with food may have limited overall absorption, and it was unknown whether patients were taking TAC under fasting conditions in this study. Finally, adherence to the prescribed TAC regimen was also not assessed.

To our knowledge, no international transplant program has evaluated TAC intragraft concentrations and their correlation with clinical outcomes. Current international literature on this topic is limited by small sample sizes, short follow-up periods, differences in immunosuppression, protocols and overall generalizability of the patient populations studied. The exclusion criteria applied to the population are TAC extended: release population, previous non-kidney transplant and missing TAC doses/troughs.

There was no correlation between same-day TAC trough in blood and kidney and liver tissue concentrations. The relationship between low TAC concentrations and BANFF score >6 was also statistically significant. The formulation of the hypothesis that low tissue concentrations of TAC may be involved in the occurrence of cellular rejections seems grounded on the basis of these early results.

Further studies are necessary in order to explain clinical utility, practical value and applicability of tacrolimus determination in kidney and liver tissue.

## Figures and Tables

**Figure 1 pharmaceutics-13-01576-f001:**
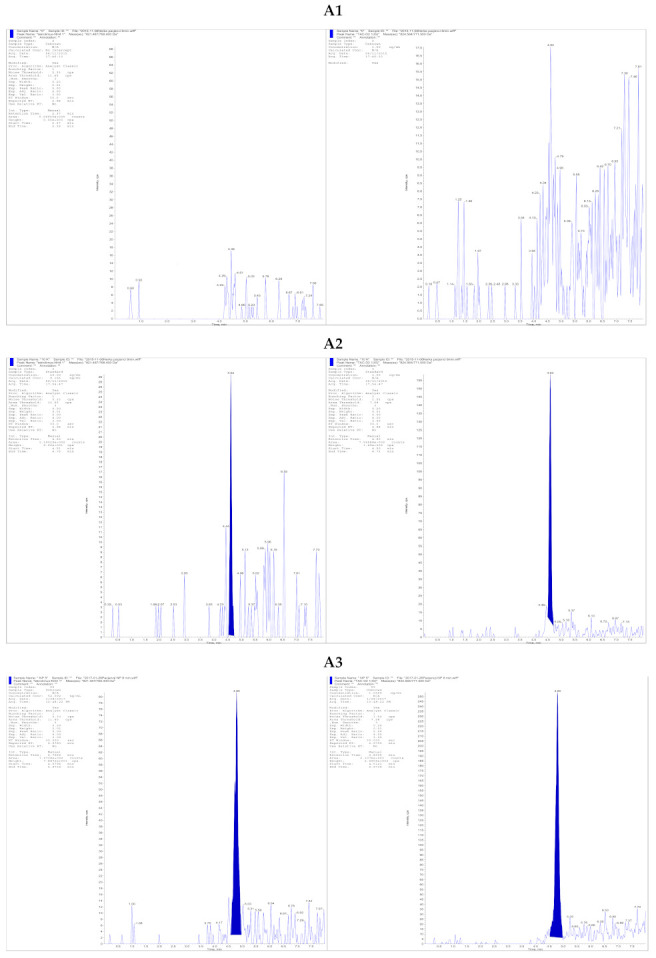
Representative multiple reaction monitoring chromatograms of (**A1**) blank kidney tissue sample without TAC and I.S. (**A2**) Tissue kidney sample containing 10 pg/mg of TAC (LOQ) with Tac^13^C-d2 (I.S.) (**A3**) Kidney tissue obtained from a patient administered with Prograf, with a measured Tac concentration of 65.1 pg/mg coeluted at 4.68 min denoted with Tac^13^C-d2 and (**B1**) blank liver tissue sample without TAC and I.S. (**B2**) Tissue liver sample containing 10 pg/mg of TAC (LOQ) with Tac^13^C-d2 (I.S.) (**B3**) A liver tissue sample obtained from a patient administered Prograf, with a measured Tac concentration of 151.4 pg/mg coeluted at 4.68 min denoted with Tac^13^C-d2.

**Figure 2 pharmaceutics-13-01576-f002:**
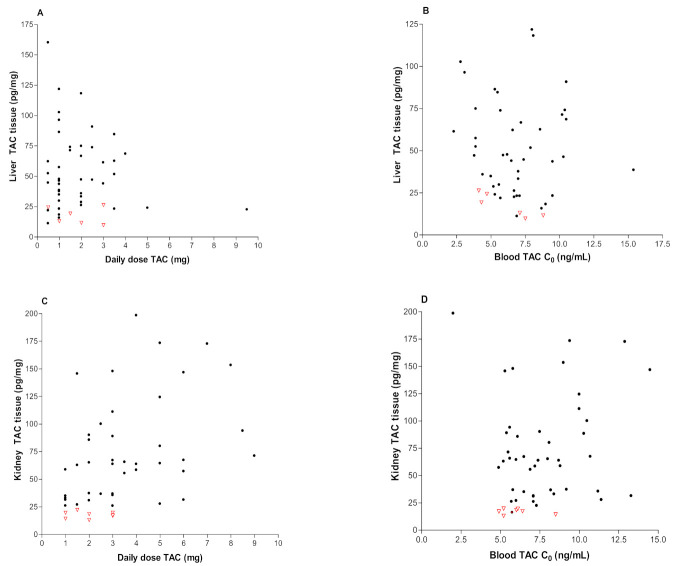
Correlations between the following: (**A**) liver TAC tissue concentrations and daily dose of Prograf (r = 0.014, *p* = 0.4244); (**B**) liver TAC tissue concentrations and blood TAC concentration (r = 0.021, *p* = 0.1519); (**C**) kidney TAC tissue concentrations and daily dose of Prograf (r = 0.073, *p* = 0.4577); and (**D**) kidney TAC tissue concentrations and blood TAC concentration (r = 0.052, *p* = 0.3918). Red “empty triangle” means recipients with moderate/severe rejection.

**Figure 3 pharmaceutics-13-01576-f003:**
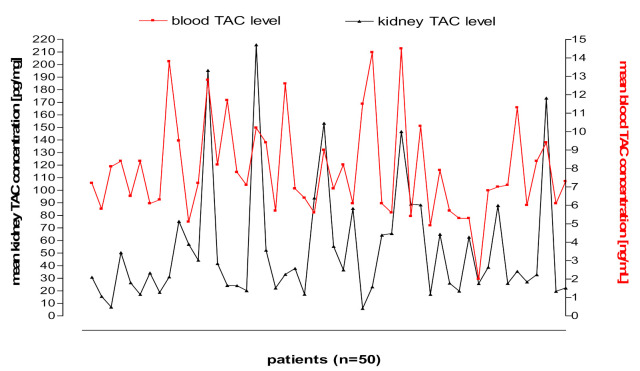
Kidney tissue and blood TAC levels determined in 50 transplant patients.

**Figure 4 pharmaceutics-13-01576-f004:**
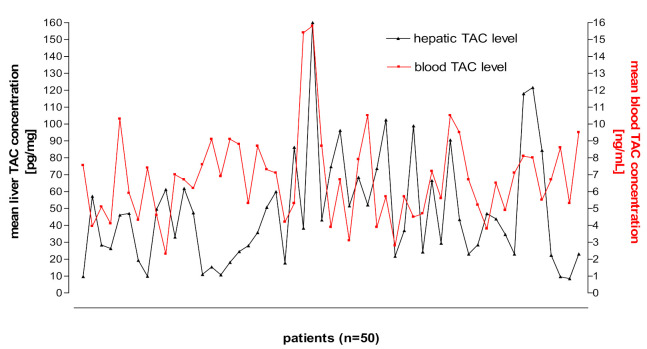
Liver tissue and blood TAC levels determined in 50 transplant patients.

**Table 1 pharmaceutics-13-01576-t001:** (**A**) Intra-assay and inter-assay precision and accuracy of LC-MS/MS method for determining TAC in kidney and liver tissues at different QC tissue concentrations. (**B**) Matrix effects in human kidney and liver samples using the postextraction analyte-addition method.

(**A**)						
**Concentration**		**Intra-Assay (n = 6)**		**Inter-Assay (n = 12)**		**Extraction** **Efficiency (%)**
**Declared (pg/mg)**	**Calculated * (pg/mg)**	**Precision CV (%)**	**Accuracy (%)**	**Precision CV (%)**	**Accuracy (%)**	
Kidney tissue						
Low QC 20	21.32 ± 1.09	6.7	102.4	9.3	111.5	75.3
Medium QC 150	158.70 ± 7.4	5.8	99.1	6.5	102.9	80.7
High QC 250	260.25 ± 12.5	4.1	96.5	7.1	105.4	88.4
TAC^13^C,D_2_ 200	204.21 ± 4.72	2.1	97.8	2.5	102.4	95.1
Liver tissue						
Low QC 20	20.86 ± 0.98	4.3	103.8	8.2	93.4	71.6
Medium QC 150	160.05 ± 11.5	6.7	105.1	9.7	107.9	81.1
High QC 250	260.55 ± 10.4	4.2	97.9	10.5	104.8	90.3
TAC^13^C,D_2_ 200	203.84 ± 4.02	1.9	98.5	9.8	99.8	96.9
LLOQ 10	10.32 ± 0.28	3.2	106.1	8.4	103.2	82.4
(**B**)						
**Type of Tissue**	**TAC Peak Area**		**TAC^13^C,D_2_ Peak Area**		**TAC/TAC^13^C,D_2_ Ratio**	
	**CV (%)**	**Bias (%)**	**CV (%)**	**Bias (%)**	**CV (%)**	**Bias (%)**
Kidney	7.4	11.2 (10 pg/mg) 9.8 (400 pg/mg)	6.3	7.8	1.8	2.4
TAC^13^C,D_2_ 400 pg/mg	5.2					
Liver	9.8	13.4 (10 pg/mg)	7.1	10.5	3.2	5.6
TAC^13^C,D_2_ 200 pg/mg	7.5					

CV, coefficient of variation; * Mean ± standard deviation.

**Table 2 pharmaceutics-13-01576-t002:** Summary of demographic and clinical parameters.

Patient Characteristics	Graft: Liver (n = 50)	Graft: Kidney (n = 50)
Age (years)	46.2 ± 13.5	46.3 ± 14.0
Gender (M/F)	33/17	40/10
ALT (IU/L)	81.3 ± 100.6	20.4 ± 16.6
AST (IU/L)	65.2 ± 71.9	19.5 ± 14.0
GGTP (IU/L)	118.4 ± 76.6	52.3 ± 19.1
GFR (mL/min/m^2^)	83.1 ± 25.7	44.7 ± 16.2
Creatinine (mg/dL)	1.01 ± 0.37	1.77 ± 0.61
Albumin (g/dL)	4.36 ± 0.49	4.21 ± 0.55
Hemoglobin (g/dL)	13.57 ± 1.78	12.1 ± 3.22
Hematocrit (%)	40.07 ± 4.68	42.2 ± 3.54
Bilirubine total (mg/dL)	1.22 ± 0.86	0.52 ± 0.28
Cold ischaemia time (h)	10.1 (4.3–12.7)	12.2 (5.7–22.5)
Acute rejection (%)	10	15
Time of allograft biopsy	30 to 180 days posttransplant	7 to 180 days posttransplant
Daily dose (mg)	1.9 ± 1.5	3.6 ± 2.4
Tissue (pg/mg)	49.9 ± 32.1	65.2 ± 50.0
Whole blood (ng/mL)	7.2 ± 3.2	7.5 ± 2.6

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
