# Peer review of "Kidney and Liver Tissue Tacrolimus Concentrations in Adult Transplant Recipients—The Influence of the Whole Blood and Tissue Concentrations on Efficiency of Treatment during Immunosuppressive Therapy"

_pharmaceutics, 2021, doi:10.3390/pharmaceutics13101576_

Round 1

Reviewer 1 Report

Thank you for the opportunity to the review this interesting manuscript by Bodnar-Broniarczyk et al. The manuscript describes the validation of an LC-MS/MS method for quantitation of tacrolimus in renal and hepatic biopsy samples, which is then applied to a clinical study of 50 kidney and 50 liver transplant recipients. There are aseveral areas that require clarification or correction before the manuscript can be accepted for publication. In particular, the biased presentation of rejection data needs to be addressed.

Abstract

The last sentence should be removed as the authors have not shown that tissue concentrations of tacrolimus were significantly lower in patients experiencing rejection. Very little numerical data are presented in the abstract.

Materials and Methods

2.1 Please provide details for the collection of blood (i.e. trough?) and allograft (drawn same time as blood or when) samples that were used to measure tacrolimus concentrations. For the allograft tissue samples, were these the same sample as used for histology assessment. How was acute rejection defined in the two transplant groups. Were subjects receiving any potentially interacting drugs?

2.3 Why were kidney and liver biopsy samples stored at different pH? How long had the patient biopsy samples been stored prior to analysis and have the authors demonstrated analyte stability over that period?

2.4 Line 118, was the ammonium acetate concentration in solvent A and B different or is there a typographic error? Capital B for solvent B. Section 2.5, how were drug-free homogenates obtained was informed consent obtained from the patients? The methods used to validate the analytical technique should be described here so the Results section only contains results.

Results

3.1 Much of the information here should be in the Materials and Methods. What was the storage stability of blank homogenates spiked with tacrolimus that were used as calibrators and QCs? In Table 1 please show extraction efficiency of tacrolimus and 13C2D-Tacrolimus separately. Please also add matrix effects data to table 1.

Tables 2 and 3 can be combined. Please check GFR values and creatinine values in Table 2 as it seems odd that liver transplants have lower GFR (poorer renal function) and lower plasma creatinine (better renal function) than renal transplants? Figures 3 and 4 could be deleted as the information in already presented in Figure 2. In Figure 2 please show the correlation coefficient and P value.

Presentation of the rejection data is very confusing and potentially biased (lines 255-268). The data in Figure 5 only show concentrations for 44/50 liver and 38/50 kidney recipients and would appear to be a biased sample. In Figure 5 the x-axis labelling is confusing and the panel for renal transplants is missing error bars for blood tacrolimus. The data seem to have been grouped according to tissue concentrations, but it is not clear why the grouping is different for the 2 panels.The SDs for blood tacrolimus in liver and renal transplants indicate that several patients in the AR groups had trough blood concentrations below the target range (<5 ng/mL) in contradiction to the statement in line 262? Line 265 is also not clear from Figure 5B. I suggest deleting Figure 5 as it is not possible to see matching blood and tissue concentrations. It would be better to show rejection as part of Figure 2 and to use different symbols (one for rejection and another for no-rejection) in Figure 2. The authors should conduct a formal statistical analysis of blood and biopsy concentrations in all subjects with and without rejection. At what time post-transplant were the episodes of rejection observed? Was rejection based only on Banff histology analysis or on clinical grounds?

Discussion

Line 276 site not side; line 299 Noll not Boll

I don’t think the authors can make any claims regarding low tissue concentrations being predictive of rejection as it appears only a selected number of biopsy/blood concentrations were presented in the data relating to rejection. In addition, the trough blood tacrolimus concentrations were not adequately displayed, particularly for the renal transplants. There were certainly patients with low tacrolimus concentrations in the groups with arrows indicating AR but it is not possible to see how the individual concentrations relate to AR or to their matching tissue concentrations. No formal statistical analysis was performed.

Author Response

To the Reviewer # 1

Thank you for reviewing the paper. We appreciate the effort you have made to help us to improve the manuscript. We carefully revised the manuscript according to the remarks. Please find below our response to each and every suggestion in a point-by-point manner

Comment: “ Abstract. The last sentence should be removed as the authors have not shown that tissue concentrations of tacrolimus were significantly lower in patients experiencing rejection. Very little numerical data are presented in the abstract”

Response: According to the reviewers` suggestion the last sentence concerning significantly lower TAC tissue concentrations in patients experiencing rejection has been removed. It`s true that it cannot be proved statistically significance TAC tissue concentrations in all patients experiencing rejection. The conclusions were reached too hastily without ROC analysis. Their results have authorized us to make the above-mentioned statement. It was just only observed in the studied group of patients that in individual cases the tissue concentrations were significantly below the average value. To sum up, we have included a new sentence summarizing the actual results.

Comment: “2.1 Please provide details for the collection of blood (i.e. trough ) and allograft (drawn same time as blood or when) samples that were used to measure tacrolimus concentrations. For the allograft tissue samples, were these the same sample as used for histology assessment. How was acute rejection defined in the two transplant groups. Were subjects receiving any potentially interacting drugs ?”

Response: Both blood and tissue samples were collected at -80°C in the deep freeze tubes. At the time of analysis, biopsies are thawed, rinsed and accurately weighted. We used a very sensitive balance to obtain accurate mass. Then specimens were homogenized using Heidolph  homogenizer in tris buffer pH 5.5 (kidney tissue) and 7.0 (liver tissue). Perhaps it was not precisely presented in the paper but graft tissue were collected without Tris- buffer. It was added (0.7 mL)  on homogenized

stage in order to later improve the sensitivity of the measured TAC at low concentrations up to 40 pg/mg. When optimizing method of TAC determination it turned out that different pH buffer used during tissue homogenization contributed to the improvement of sensitivity. Appropriate changes to the text of the section 2.3 have been made.

Blood samples were drawn and biopsies were performed on the same day, namely blood sample at 7.00 a.m. biopsy 10.00 a.m. – 11.00 a.m. For the allograft tissue samples, both kidney and liver, were the same sample  as used for histology assessment. Acute rejection was graded in case of kidney biopsy according to the  BANFF score, in case of  liver biopsy according to the RAI index ( Rejection Activity Index). The subject were not receiving potentially interacting drugs.

Comment: “2.3 Why were kidney and liver biopsy samples stored at different pH?  How long had the patient  biopsy samples  been stored prior to analysis and have the authors demonstrated analyte stability over that period ?”

Response: Partially it was explained the different of pH in response for comments 2.1 ( above). The patient biopsy samples were stored max. 1 month before analysis. According to the earlier studies sample stability of tacrolimus in blood and some alternative matrices were investigated and presented f.e. in blood  for 1 month at -20°C and 1 year at -70°C ( Dubbelboer IR et al., Ther Drug Monit., 2012; 34: 134-142; Freeman DJ et al. Ther Drug Monit 1995; 17: 266-267) and in alternative matrices f.e. PBMC for  up to 3 months at -30°C, in human bile samples for 6 months at -80°C (Romano P et al. J Pharm Biomed Anal. 2018; 152: 306-314  ; Tron C. et al. J. Chromatogr A. 2016; 1475: 55-63). Therefore, we have not demonstrated samples stability within 6 months, especially we often obtained  a very small amount of tissue , about 1 mg.

Comment “2.4 Line 118, was the ammonium acetate concentration in solvent A and B different or is the typographic error ? Capital B for solvent B.”

Response: We highly appreciate the remark given. It was a mistake. There is a typographic error. The ammonium acetate concentration in both solvents: A and B is the same: 2.5 mmol/L. Capital B instead of lowercase b. It has been corrected.

Comment “2.4:The methods used to validate the analytical technique should be described here.”

Response: We absolutely agree with the Reviewer. Therefore section Material and Methods have been supplemented with relevant sub-chapters and transferred from section results.

Comment: “2.5 How were drug-free homogenates obtained was informed consent obtained from the patients ? The methods used to validate the analytical technique should be described here so the Results section only contains results.”

Response: Drug-free homogenates were obtained from human kidney in conjunction with nephrectomy due to neoplastic changes (tumor). Drug-free homogenates were obtained from human liver taken from patients with indication for transplantation due to HCC (hepatocellular carcinoma) without cirrhosis. Patients had never been treated with TAC. The method development protocol and the clinical trial protocol and evaluated and approved by the Regional Bioethical Committee and performed in accordance with the Declaration of Helsinki. All patients signed a written informed consent.

Comment: “3.1. Much of the information here should be in the Materials and Methods. What was the storage stability of blank homogenates spiked with tacrolimus that were used as calibrators and QCs ?”

Response: The storage stability of blank homogenates spiked with tacrolimus that were used as calibrators has not been tested. It was because that just after preparing, calibrators within 24 hours were used.

Comment: “In Table 1 please show extraction efficiency of tacrolimus and 13C2D-Tacrolimus separately. Please also add matrix effects data to Table 1. Table 2 and 3 can be combined.”

Response: According to Reviewers` suggestion   Table 1 presents extraction efficiency both for Tacrolimus and deuterated internal standard. In the postextraction analyte-addition study, no significant matrix effects were observed for  2 different ( low 20 pg/mg and high 250 pg/mg concentration) with regards to final TAC concentrations (TAC/I.S. area ratio bias and CV<15%, Table 1). Postcolumn infusion experiments identified that TAC and TAC13Cd2-TAC coeluted in an area of nonsignificant  ion enhancement caused by the extracted transplant kidney and liver biopsy tissue.

Table 2 and 3 is combined.

Comment: “Please check GFR values and creatinine values in Table 2 as it seems odd that liver transplants have lower GFR (poorer renal function) and lower plasma creatinine (better renal function) than renal transplants ?”

Response: I apologize for the errors that occurred in the GFR calculations in case of liver transplant patients. It was due to incorrect data entry in the calculation of the GFR. Of course,  liver transplant have higher  GFR ( better renal function) than renal transplant.  In case of liver transplant patients mean ± SD of GFR should be 83.1 ± 25.7 ml/min/m2 instead of 21.6 ± 8.4 ml/min/m2.  It should not  happen. Comment: “Figures 3 and 4 could be deleted as the information is already presented in  Figure 2. In Figure 2 please show the correlation coefficient and P value.” Response: The authors propose to keep two figures: 3 and 4 as simultaneous visualization of TAC concentration in blood and tissue transplant patients is very legible on them. The correlation coefficient and P value are presented in Figure 2. Comment: “Presentation of the rejection data  is very confusing and potentially biased (lines 255-268). The data in Figure 5 only show concentrations for 44/50 liver and 38/50 kidney recipients and would appear  to be a biased sample. The data seem to have been grouped according to tissue concentrations, but it is not clear why the grouping is different for the 2 panels. I suggest deleting Figure 5 as it is not possible to see matching blood and tissue concentrations. It would be better to show rejection as part of Figure 2 and to use different symbols (one for rejection and another for no-rejection) in Figure 2, The authors should conduct a formal statistical analysis of blood and biopsy concentrations in all subjects with and without rejection.”Response: The results presented in Figure 5 may appear discrepancies. However the goal for the authors was to show differences between TAC concentration in several selected groups of patients with or without acute rejection. We tried to avoid any manipulation, but only to show the reader some differences in concentration value. Hence, some patients, whose concentration values ​​did not influence the observed phenomena, were not taken into account. Taking into account the Reviewer's request, we see only two ways of solving the problem: omission from figure 5 and consideration of the results in figure 2 or supplementing figure 5 with patients without concentration levels significant for the discussed problem. Of course we conduct a formal statistical analysis of blood and biopsy TAC concentration for two groups of patients with or without rejection and present them separately.  

Comment: “At what time post-transplant were the episodes of rejection observed ? Was rejection based only on Banff histology analysis or on clinical grounds ?”

Response: The episodes of rejection were observed up to 6 months post-transplantation. Procedural and indication biopsy were confirmed. Rejection was based on Banff histology analysis and Rejection Activity Index. Relevant information on this subject was included in the paper. 

Comment: “Line 276 site not side ; line 299 Noll not Boll”

Response: The correction has been effected.

Comment: “I don`t think  the authors can make any claims regarding low tissue concentrations being predictive of rejection as it appears only a selected number of biopsy/blood concentrations were presented in the data relating to rejection. There were certainly patients with low tacrolimus concentrations in the groups with arrows indicating AR but it is not possible to see how the individual concentrations relate to AR or to their matching tissue concentrations.”

Response: Thank you for your comment. This is noted and appreciated.  It was an unauthorized request that has been completely changed. This conclusion could not be drawn without ROC curve analysis, which  was impossible to use in the study due to insufficient data. The authors wanted only to draw attention to the fact that low level of TAC concentration in graft tissue could contribute to graft rejection.

Therefore, the conclusions have been refined to the actual results and further investigation should be conducted to confirm the validity of the observed phenomena.

Reviewer 2 Report

I read with interest the manuscript written by Bodnar-Broniarczyk and colleagues. The authors present a study conducted in liver and kidney transplant recipients aiming at evaluating the relationship between blood and tissue concentrations of tacrolimus in solid organ transplant recipients.

Here are my comments on the manuscript:

General comment: The manuscript would benefit from a comprehensive edition by an English native speaker. There are some spelling errors, which should be corrected such as: line 41 “in the low temperature” to “at low temperature”, line 65 “a better understanding a mechanism” to “a better understanding of the mechanism”; line 117 “a mixture of a solvents” to “a mixture of solvents”; line 118 “solvent b” to “solvent B”; tacrolimus is sometimes written as an acronym and sometimes not etc… The authors should carefully reviewed their manuscript and the final version should be edited by a professional or native-speaker.

Major concerns:

Introduction:

1/ Some parts of the introduction do not perfectly connected. The authors first stated that tacrolimus trough concentration is correlated with the drug exposure and can be used as a surrogate for AUC before saying that trough does not perfectly correlate to AUC. Then, they use this statement as a point to dismiss trough concentration as a surrogate for tissue concentration which is not supported by the previous statement. Finally, they connected this part with a statement on the narrow therapeutic index of the drug and the need for drug dosage individualization. The whole part has to be rewritten. Maybe, they should state that troughs can be usually used as a surrogate for exposure but that tissue distribution can be influenced by factors disturbing the extracellular-intracellular equilibrium potentially leading to unpredictable tissue concentrations. This change in tissue concentrations might then preclude ensuring drug efficacy and adverse drug reaction onset.

2/ Lines 45-46: The authors should provide a reference for the sentence: “TAC exhibits wide inter- as well as intra- individual pharmacokinetics variability in both kidney and especially liver transplant recipients”

Materials and Methods:

3/ A part on how tissues have be obtained and prepared should be added in this section.

4/ Lines 98-99: Why did the authors used different pH for their buffer according to the type of tissue (Kidney or Liver)?

5/ Figure 1: It is unclear what is part A and what is part B because there is no description on the part B in the figure legend. The authors should show the chromatogram of a black plasma and the LOQ as the background noise seems to be important. Plus, why are the retention times different between figure A and figure B?

6/ Precision and accuracy tests should have been presented for the QCs levels (20, 150, 250 pg/mg) not for the calibration concentration points.

Results:

7/ Line 188: Is it 10 and 400 pg/mg or pg/mL?

8/ Again, line 196: is it 10 pg/mg?

9/ Regarding acute rejection, was the biopsies conducted systematically? If yes, 10% appears relatively low in liver transplantation while 15% appears relatively high in kidney transplantation for biopsies within the first post-operative 6 months. This comment is related to the lack of explanation of how the authors obtained the tissues.

10/ How were the biopsies performed regarding the time of treatment intake? This should be stated as it is possibly of importance regarding the correlation between trough concentrations and biopsies performed at any time during the administration interval.

11/ May the authors provided the correlation coefficient and the degree of significance for all their figures?

12/ I don’t understand why in figure 5, only 9 and 4 patients’ anatomopathological results were displayed for liver and kidney recipients, respectively. If all the patients benefited from a biopsy, the authors should present all the results. Please comment.

Discussion:

13/ Line 276-277: why do the authors it would be more interesting to evaluate tissue concentrations than lymphocyte, which is the real site of action of tacrolimus, concentration?

14/ Lines 277-282: The authors should maybe add that obtaining tissues using biopsy is a very invasive procedure with a non-negligible morbidity.

15/ The authors should maybe discussed the interest of intracellular concentration measurement as a surrogate for tissue concentration as highlighted in the work by Capron and colleagues (Ref #22). Biopsies is an invasive approach and a proxy of tissue concentration measurement is an objective in solid organ transplantation.

16/ What about peri-tissular tacrolimus concentration measurement? Direct effluents like urine for kidney or bile for liver might reflect the graft tissue concentration. Is it a potentially useful approach?

17/ How did the authors end with the conclusion of a 19-21 pg/mg threshold for acute rejection prevention? There is no such mention of the threshold in the results section of the manuscript and no sensitivity analysis aiming at determining a threshold is proposed. This point should absolutely be clarified.

Minor comments:

18/ In the abstract, “side effect” should be changed to the more precise medical wording “adverse drug reaction”.

19/ Table 1: There is a dot after determining that should be deleted.

20/ Table 2: Acronyms should be defined.

21/ Line 231: “55,9” has to be changed to “55.9”

22/ Figure 2 legend. There is an upper case line 245 “And” and line 246 “And”.

23/ Line 276: Please change “side” for “site”

24/ Line 280: A dot is missing after “contamination”.

25/ Line 321: ABCB1 and CYP3A5 in italic.

Author Response

To the Reviewer # 2

Comment: “I read with interest the manuscript written by Bodnar-Broniarczyk and colleagues.”

Response: Thank you for reviewing the paper. We appreciate the effort you have made to help us to improve the manuscript.. Thank you for your helpful suggestions. We agree and we made the necessary changes. We carefully revised the manuscript according to the remarks. Please find below our response to each and every suggestion in a point-by-point manner.

“Here are my comments on the manuscript:”

General comment:  “The manuscript would benefit from a comprehensive edition by an English native speaker. The authors should carefully reviewed their manuscript and the final version should be edited by a professional or native –speaker.”

Response: The authors carefully reviewed their manuscript and the re-submitted version of the manuscript have  been edited by a native-speaker. We apologize  for the linguistic deficiencies that occurred in the original version of the work.

Comment: Major concerns

Introduction

Comment: “1/ Some parts of the introduction do not perfectly connected. The authors first stated that tacrolimus trough concentration is correlated with the drug exposure and can be used as a surrogate for AUC before saying that trough does not perfectly correlate to AUC. Then, they use this statement as a point to dismiss trough concentration as a surrogate for tissue concentration which is not supported by the previous statement. Finally, they connected this part with a statement on the narrow therapeutic index of the drug and the need for drug dosage individualization. The whole  part has to be rewritten.”

Response: We absolutely agree with the Reviewer. According to Reviewers` suggestion some parts of Introduction have been changed and made more consistent from the logical point of view

Comment: “2/ Line 45-46: The authors should provide a reference for the sentence: “ TAC exhibits wide inter- as well as intra- individual PKs variability in both kidney and especially liver transplant recipients”

Response: Thank you for pointing this out. We provide a reference for the sentence: Campagne O., Mager D.E., Tornatore K.H. Population Pharmacokinetics of Tacrolimus in Transplant Recipients: What did we learn about sources of interindividual variabilites ? J Clin Pharmacol , 59(3), 309-325, Trull A.K. Therapeutic monitoring of tacrolimus. Ann Clin Biochem 1998, 35, 167-180.

Materials and Methods:

Comment: “3/  A part on how tissues have been obtained and prepared should be added in this section.”

Response: We acknowledge the importance of establishing the conditions of tissue sampling and preparing to analysis. Similarly to the answer given to the reviewer #2 I present the method and conditions below.

Blood samples were drawn and biopsies were performed on the same day, namely blood sample at 7.00 a.m. biopsy 10.00 – 11.00 a.m. For the allograft tissue samples, both kidney and liver, were the same sample  as used for histology assessment. Acute rejection was graded in case of kidney biopsy according to the  BANFF score, in case of  liver biopsy according to the RAI index ( Rejection Activity Index).

Both blood and tissue samples were collected at -80°C in the deep freeze tubes. At the time of analysis, biopsies are thawed, rinse and accurately weighted. We used a very sensitive  balance to obtained accurate mass. Then specimens were homogenized using Heidolph  homogenizer in tris buffer pH 5.5 ( kidney tissue) and 7.0 ( liver tissue). Perhaps it was not precisely presented in the paper, but graft tissue were collected without Tris- buffer. Tris buffer (0.7 mL) was added on homogenized stage in order to later improve the sensitivity of the measured TAC at low concentrations up to 40 pg / mg. When optimizing the method of determination of TAC it turned out that different pH buffer used during tissue homogenization contributed to the improvement of sensitivity.

Comment: “4/ Lines 98-99: Why did the authors used different pH for their buffer according to the type of tissue (Kidney or Liver) ?”

Response: Thank you for pointing out this obvious error, which has been corrected appropriately. The answer has been included in response to a previous comment.

Comment: “5/  Figure 1: It is unclear what is part A and what is part B because there is no description on the part B in the figure legend. The authors should show the chromatogram of a blank plasma and the LOQ as the background noise seems to be important. Plus, why the retention times different between figure A and figure B.”

Response: We really appreciate for the missing capital letter B in description Figure 1. It has been completed. We have also completed chromatograms regarding presentation of blank plasma without drug and LOQ. Minimal differences in retention times not exceeding 0.04 min ( 2.4 s) arise from this that analysis was carried out on different days and slight variation in parameters during analysis worked in minimal differences in RT.

Comment: “6/ Precision and accuracy tests  should have been presented for the QCs levels (20, 150, 250 pg/mg) not for calibration concentration points.”

Response: In line with the Reviewers` comments precision and accuracy is presented for the QCs levels instead of the calibration concentration points.in Table 1. Moreover matrix effect data are additionally included in this table.

Results:

Comment: “7/ Line 188: Is it 10 and 400 pg/mg or pg/mL ?”

Response: Of course it should be pg/mg. The unit has been improved in the text.

Comment: “8/ Again, line 196: is it 10 pg/mg ?”

Response: Yes , you are right. Similarly to the note above.

Comment: “9/ Regarding acute rejection, was the biopsies conducted systematically ? If yes, 10 % appears relatively low in liver transplantation while 15 % appears relatively high in kidney transplantation for biopsies within the first post-operative 6 months. This comment is related to the lack of explanation of how the authors obtained the tissues.”

Response: Biopsies were systematically performed before noon (10.00 – 11.00 a.m.) most of as protocol biopsies and indication biopsies. According to the authors knowledge and experience 10 % of acute rejection for liver transplantation is not to low in case of protocol biopsies, the more that it was taken into account  as clinically significant organ rejection according to RAI >6. In reference to the comment that 15 % appears relatively high in kidney transplantation for biopsies within the first post-operative 6 months it seems to us that taking into account protocol and indication biopsies the results are rather normal.

Comment: “10/ How were the biopsies performed regarding the time of treatment intake ? This should be stated as it is possibly of importance regarding the correlation between trough concentrations and biopsies performed at any time  during the administration interval.”

Response: The drugs: tacrolimus (Prograf), mycophenolate mofetil, and glucocorticosteroids were administered  at 7 a.m. Biopsies were performed between 10.00 and 11.00 a.m. So in this particular case the time difference between the drug taken and the biopsy performed is 3-4 hours. It seems impossible to perform a biopsy before starting the administration interval. The rules of the clinic do not allow this.

Comment: “11/ May the authors provided the correlation coefficient and the degree of significance for all their figures ?”

Response: Correlation figures were supplemented by data: correlation coefficient and   P value.

Comment “12/  I don`t understand why in figure 5, only 9 and 4 patients` anatomopathological results were displayed for liver and kidney recipients, respectively. If all the patients benefited from a biopsy, the authors should present all the results. Please comment.”

Response: Similarly to the response for comment from Reviewer #2  I would like to explain the reason why Figure 5 show partial results. The results presented in Figure 5 may appear discrepancies. However the goal for the authors was to show differences between TAC concentration in several selected groups of patients with or without acute rejection. We tried to avoid any manipulation, but only to show the reader some differences in concentration value.

 Hence, some patients, whose concentration values ​​did not influence the observed phenomena, were not taken into account. Taking into account the Reviewer's request, we see only two ways of solving the problem: omission from figure 5 and consideration of the results in figure 2 or supplementing figure 5 with patients without concentration levels significant for the discussed problem. Of course we conduct a formal statistical analysis of blood and biopsy TAC concentration for two groups of patients with or without rejection and present separately.

Discussion:

Comment:”13/ Line 276-277 why do the authors it would be more interesting to evaluate tissue concentrations than lymphocyte, which is the real site of action of tacrolimus concentration ?”

Response: It is true. Determining tacrolimus concentrations where it exerts its immunosuppre-ssive effect, in the T cell or in PBMC might be particularly relevant. On the other hand, measuring tacrolimus directly in graft tissue may be of interest and it is reasonable to expect that local concentrations better reflect drug effect. A study  confirming the link between intracellular tacrolimus concentration and patient outcomes after liver transplantation suggested that this new Therapeutic Drug Monitoring (Ref #22) approach was a valuable option,  but definitive clinical verification and validation remain to be generated. Moreover in that case daily TAC monitoring was performed using CMIA on the ARCHITECT platform and Tacrolimus measurement in tissue was performed by LC-MS/MS.  The intrahepatic and intrarenal  peripheral blood mononuclear cells measurement of tacrolimus levels are more complicated and we did not have the possibility to carry out such research at that time. A more recent paper ( Ref #24: Capron A et al. Pharmacol. Res.  2016, 111:610-618) confirms that intra-cellular immunosuppressive drug monitoring states an additional tool to individualize more precisely early immunosuppressive schemes after liver transplantation.

Comment: “14/ Lines 277-282: The authors should maybe add that obtaining tissues using biopsy is a very invasive procedure with a non-negligible morbidity.”

Response: We absolutely agree with the reviewer that obtaining tissues using protocol biopsy is an invasive procedure with a non-negligible morbidity. At the same time, it is a procedure that leaves no significant complications and is commonly used in the transplantation clinic. The relevant sentence was included in the text.

Comment: “15/ The authors should maybe  discussed the interest of intracellular concentration measurement as a surrogate for tissue concentration as highlighted in the work by Capron and colleagues (Ref #22). Biopsies is an invasive approach and a proxy of tissue concentration measurement is an objective in solid organ transplantation.”

Response: We highly appreciate the indication given. I have referred to the above-mentioned comment in the above two responses. An appropriate comment has been posted in the Discussion section.

Comment: “16/  What about  peri-tissular tacrolimus concentration measurement ? Direct effluents like urine for kidney or bile for liver might reflect the graft tissue concentration. Is it potentially useful approach ?”

Response: I don`t think so. Admittedly in case of liver transplant recipients one publication exists concerning biliary tacrolimus concentration measurement using LC-MS/MS as a new pharmacological marker of immunosuppressive activity ( Tron C et al. J. Chromatogr A, 2016, 1475: 55-63), but in our opinion ECPW (retrograde cholangiopancreatography) is also a highly invasive procedure. It is true  that tacrolimus is metabolized in the liver and eliminated through biliary excretion. Therefore there is a hypothesis that tacrolimus concentration measured in excreted bile could be a relevant surrogate marker of its efficacy. Preliminary data about tacrolimus excretion profile in bile showed the lack of correlation between tacrolimus whole blood concentration and tacrolimus liver  exposition. Moreover, this procedure could be at risk of acute pancreatitis.In relation to kidney transplant recipients we know that just only 2% of the tacrolimus is eliminated in urine. In this case, monitoring using urine as a matrix is not relevant and is not practiced. 

Comment: “17/ How did the authors end with the conclusion of a 19-21 pg/mg threshold for acute rejection prevention ? There is no such  mention of the threshold in the results section of the manuscript and no sensitivity analysis aiming at determining a threshold is proposed. This point should absolutely be clarified.”

Response: You are absolutely right. It was an unauthorized request that has been completely changed. This conclusion could not be drawn without ROC curve analysis, which was impossible to use in the study due to insufficient data. The authors wanted only to draw attention to the fact, that low level of TAC concentration in graft tissue could contribute to graft rejection.

Therefore, the conclusions have been refined to the actual results and further investigation is recommended to be conducted to confirm the validity of the observed phenomena.

Minor comments:

Comments:

18/ In the abstract, “side effect” should be changed to the more precise medical wording “adverse drug reaction”.

19/ Table 1: There is a dot after determining that should be deleted.

20/ Table 2 Acronyms should be defined.

21/ Line 231: “55,9” has to be changed to “55.9”.

22/ Figure 2 legend. There is an upper case line 245 “And”  and line 246 “And”.

23/ Line 276: Please change “side” for “site”.

24/ Line 280: A dot is missing after “contamination”.

25/ Line 321: ABCB1 and CYP3A5 in italic.

Response: All the remarks have been attended to.

Reviewer 3 Report

Previous some studies indicate that inter-tissue TAC concentration can be better predictor in regard to acute rejection episode than TAC concentration in whole blood. This study showed that intra-tissue concentrations of TAC were significantly lower in kidney and liver transplant recipients experiencing rejection than those who did not ,although TAC concentrations in whole blood were maintained in therapeutic range. This study is informative and useful for transplant physicians.

There was no correlation between dosage, blood concentrations and tissue TAC concentrations. Please explain the possible mechanism or reason in the Discussion section.

Author Response

To the Reviewer #3:

Comment: “ Previous some studies indicate that inter-tissue TAC concentration can be better predictor in regard to acute rejection episode than TAC concentration in whole blood. This study showed that intra-tissue concentrations of TAC were significantly lower in kidney and liver transplant recipients experiencing rejection than those who did not, although TAC concentrations in whole blood were maintained in therapeutic range. This study is informative and useful for transplant physicians.”

Response: Thank you for reviewing the paper. We appreciate the effort  you have made to help us to improve the manuscript.

Comment: “There was no correlation between dosage, blood concentrations and tissue TAC concentrations. Please explain the possible mechanism or reason in the Discussion section.”

Response: I really appreciate this suggestion. The hypothesis is that significant variability of tacrolimus concentrations may lead alternatively to underexposure and overexposure periods resulting in immune activations with subclinical rejections accumulation favouring organ lesions and drug toxicity associated with adverse events and organ damages. According to current knowledge about some aspects of pharmacogenetics during immunosuppressive therapy the  association  between CYP3A5 genotype and tacrolimus dose requirements  is consistent and has been observed among kidney and liver transplant recipients. Patients expressing CYP3A5 require at least 50 % higher tacrolimus dose to reach the target therapeutic range compared with non-expressors. I am sorry to say that among limitations of the study, there were no PG investigations regarding PG polymorphism. On the other hand, distribution of TAC to transplanted tissue ( the tissue drug level) would be closely modulated by the expression and polymorphisms of drug transporters f. e.  MDR1 polymorphisms could be partly responsible for the wide range of TAC concentrations observed in graft tissue. Also because of the time difference between the administered drug and the biopsy performed (3-4 hours) correlation between dose and tissue concentration is limited. A complex mechanism is advised to be investigated to explain the real relationship between  dosage and TAC blood and tissue concentration.

Round 2

Reviewer 1 Report

I would like to thank the authors for their considered responses to my comments. For the most part they have addressed many of my concerns. However, there are still issues that should be cleared up before the manuscript is suitable for publication. A few other confusing areas are also apparent.

Major concerns:

Lines 125-126, please state how whole blood tacrolimus concentrations were measured. A simple citation of a published method would be sufficient.

Lines 217-222, given that in the methods the authors indicate that calibration curves were prepared over a range of protein concentrations, could the authors please add one or two sentences to indicate whether the protein amount affected linearity and what amount of blank homogenate protein was finally chosen for routine calibration, QCs and determination of accuracy, reproducibility and matrix effects.

Lines 226-233 the information in the text is different to that in Table 1. Could the authors please clarify which data apply to calibrators and which to QCs.

Line 274, data should be shown for all patients not for “selected” patients. Do the authors mean those with moderate/severe rejection?
Line 277, “with no or mild cellular rejection…”
Line 279 please give P value for comparison of renal tissue concentrations between no/mild and moderate/severe rejection.
Line 283 please add “(n=)” value after pg/mg
Line 284 please also add “(n=)” value after pg/mg, should beginning of this line be (RAI 0-5)? Please add P value for comparison of liver tac concentrations between no/mild and moderate/severe rejection.
Line 286, if they are statistically significant, please add P value for comparison of blood concentrations

Figure 2: please label all panels (A, B, C, D), and increase the size of the dots. In the graphing software used to create the two right-hand side panels, is it possible to use two separate symbols? One for the no/mild and another for moderate/severe? Hand drawn circles are not suitable. Also why does the kidney Tac panel only show 3 circles for moderate/severe rejection when the text indicates (n=7)? Please show all the data. Please amend legend using (A, B, C, D) and no/mild or moderate/severe (as appropriate for rejection status). It would be interesting to show the patients with rejection in all four panels.

The discussion should be more succinct and should be reviewed for English language usage by a professional (scientist) native speaker.
Lines 334-337, Please delete “The significant variability …. events and organ damages”. This does not address lack of correlation.
Lines 337-341 I assume these sentences are highlighting one of the reasons for the lack of correlation between dose and concentration. Perhaps authors could briefly state this after “non-expressors [3].” Similarly, after “… graft tissue” on line 344 authors could briefly state this may contribute to lack of correlation between blood and tissue concentrations?
Line 343 please use ABCB1 (not MDR1).

Lines 366-394 could be significantly shortened. I’m not sure of the relevance of lines 370-381. They could be deleted.

Other comments:

Line 86, include formulation name for mycophenolate
Section 2.4 when was internal standard added?
Line 174, also include post-extraction addition
line 175, pg/?
Table 1B what concentrations of TAC and IS were used?
Line 269, Table 2
Line 272, liver and kidney transplant recipients
